

# Ice shelf rift propagation: stability, three dimensional effects, and the role of marginal weakening

Bradley Paul Lipovsky

Department of Earth and Planetary Sciences
Harvard University

**Correspondence:** brad_lipovsky@fas.harvard.edu

**Abstract.** Understanding the processes that govern ice shelf extent are of fundamental importance to improved estimates of future sea level rise. In present-day Antarctica, ice shelf extent is most commonly determined by the propagation of through-cutting fractures called ice shelf rifts. Here, I present the first three-dimensional analysis of ice shelf rift propagation. I present a linear elastic fracture mechanical (LEFM) description of rift propagation. The model predicts that rifts may be stabilized when buoyant flexure results in contact at the tops of the near-tip rift walls. This stabilizing tendency may be overcome, however, by processes that act in the ice shelf margins. In particular, both marginal weakening and the advection of rifts into an ice tongue are shown to be processes that may trigger rift propagation. Marginal shear stress is shown to be the determining factor that governs these types of rift instability. I furthermore show that rift stability is closely related to the transition from uniaxial to biaxial extension known as the compressive arch. Although the partial contact of rift walls is fundamentally a three-dimensional process, I demonstrate that it may be parameterized within more numerically efficient two-dimensional calculations. This study provides a step towards a description of calving physics that is based in fracture mechanics.



## 1 Introduction

The Antarctic ice sheet is projected to loose mass this century. Although the rates of mass loss over this timeframe are typically

projected to mirror recent rates, several types of more extreme ice sheet response to global climate forcing cannot presently be excluded (Pattyn et al., 2018). Perhaps the most prominent of these extreme changes is the retreat of the floating ice shelves that fringe the Antarctic continent. Ice shelf retreat has been observed to occur gradually, i.e., over a period of years to decades (MacGregor et al., 2012; Arndt et al., 2018), and also abruptly, i.e., over a period of weeks to months (Scambos et al., 2000; Banwell et al., 2013). Although ice shelves themselves do not contribute to sea level rise, they do act to buttress grounded ice

(Rignot et al., 2004; Scambos et al., 2004; Goldberg et al., 2009; Gudmundsson, 2013). For this reason, ice sheet mass and therefore global mean sea level are closely connected to the extent and stability of ice shelves. Here, I examine the stability of ice shelves with respect to the propagation of large through-cutting fractures called rifts.

The largest modern ice shelves exist in embayments. This basic observation has long prompted the notion that embayments promote the existence of large stable ice shelves (Hughes, 1977; Thomas and Bentley, 1978; Sanderson, 1979; Rist et al.,

2002). Yet not all ice shelves fully fill the largest possible embayment. The Pine Island Glacier Ice Shelf, for example, does not presently fill the entire embayment between Bear Peninsula and Thurston Island; instead it fills the much smaller local embayment of Pine Island Bay. Furthermore, analysis of sediment cores (Naish et al., 2009) and iceberg scour marks (Yokoyama et al., 2016) suggest that past ice shelves have waxed and waned in extent through ice age cycles. Although embayments appear to stabilize ice shelves, it would therefore appear that some other process is responsible for determining the size of a stable

ice shelf within a given coastal geometry. The close relationship between the state of stress in an ice shelf and the ice shelf boundary conditions (Budd, 1966; MacAyeal, 1989) motivates investigation into processes acting in ice shelf margins.

Ice shelf margins are the region of the ice shelf grounding zone that is roughly parallel to flow (see Fig. 1). The importance of ice shelf margins is suggested by several observations, foremost among these being the observation of marginal weakening prior to ice shelf collapse. Estimates of ice rheology based on the inversion of surface velocity fields show extensive marginal

weakening prior to the collapse of the Larsen A (Doake et al., 1998) and Larsen B Ice Shelves (Vieli et al., 2006; Khazendar et al., 2007). Although ice shelf collapse (i.e., total and rapid retreat) is a complex phenomenon that involves other processes besides rift propagation (Banwell et al., 2013), rift propagation does appear to play a role in collapse. Glasser and Scambos (2008) explicitly noted that marginal weakening immediately preceded rift propagation and eventual collapse on Larsen B. Further observation of a relationship between ice shelf retreat, rifting, and marginal thinning has been noted in the Amundsen

Sea Embayment (MacGregor et al., 2012) and Jakobshavn Isbrae, Greenland (Joughin et al., 2008). Motivated by these observations, a central question of this paper is, what is the precise mechanical relationship between ice shelf margins and ice shelf rift propagation?

The main result of this paper is that marginal weakening can destabilize rift propagation. I begin by providing background on the state of stress in an ice shelf as well as some aspects of linear elastic fracture mechanics (LEFM) in Sections 2 and 3. A

more precise statement of the main result is then given in Section 4, where I also examine a simplified analytical treatment of



the three-dimensional calculations. I conclude by discussing the relationship between rift propagation, the compressive arch, rift-filling melange, and ocean swell in Section 5.

## 2   Background

I consider an ice shelf to be a buoyantly floating elastic plate of uniform thickness. As such, a seaward-facing ice front experi-
ences both a net bending moment and an in-plane horizontal membrane stress (Weertman, 1957; Reeh, 1968). The vertically-
averaged membrane stress is,

$$\sigma_m \equiv \frac{\rho g h}{2}\left(1 - \frac{\rho}{\rho_w}\right) \tag{1}$$

while the bending moment is given by,

$$m_0 \quad \equiv \quad \frac{\rho g h^3}{12}\left[3\left(\frac{\rho}{\rho_w}\right) - 2\left(\frac{\rho}{\rho_w}\right)^2 - 1\right] \equiv \phi\frac{\rho g h^3}{12}. \tag{2}$$

In these expressions, $\rho$ and $\rho_w$ are the densities of ice and water and $h$ is the ice thickness. Typical values of $\rho/\rho_w = 0.90$ give
$\phi = 0.08$. The bending moment may also be expressed as a bending stress,

$$\sigma_b \equiv \frac{6m_0}{h^2} = \phi\frac{\rho g h}{2}. \tag{3}$$

The bending stress $\sigma_b$ is the value of the rift-normal stress at the top of the ice shelf; it is also the maximum value of the
rift-normal stress. The horizontal component of loading (Eq. 1) is commonly used as a boundary condition in numerical ice
flow models, whereas the bending moment is not typically applied in ice sheet models because its effects are confined to a
narrow boundary layer in the vicinity of the ice front (MacAyeal, 1989).

Rifts walls have the same ice-front boundary conditions as a seaward-facing ice front. The main difference between a
seaward-facing ice front and a rift wall is that it is possible for rift walls to come into contact. This contact is expected to occur
at the top of the ice shelf and in the region near the rift tip, as illustrated in Fig. 1b. Indeed, De Rydt et al. (2018) recently
observed that a rift tip on the Brunt Ice Shelf was further advanced at depth than at the surface, suggesting the occurrence of
partial contact. I examine the partial contact of rift walls in Section 4. As an aspect of linear elastic fracture mechanics, fracture
wall contact is a well-studied topic (Tada et al., 2000, Chapter 1, Part C).

I use full three-dimensional elasticity calculations combined with linear elastic fracture mechanics (LEFM) to examine the
propagation of ice shelf rifts. Although a number of previous studies have examined ice shelf rifts using LEFM, no previous
study appears to have considered three-dimensional effects. Hulbe et al. (2010) calculated two-dimensional mixed mode (in-
plane opening and shearing) stress intensity factors and as a result was able to state a fracture condition as well as predict rift
propagation paths. Other ice shelf LEFM studies have mostly focused on propagation paths (Plate et al., 2012; Levermann
et al., 2012; Borstad et al., 2017) and near-tip deformation (Larour et al., 2004a, b).

A final point of background concerns the relationship between the forces that drive fracture and the background ice flow.
In real-world ice shelves, the state of stress is constantly evolving due to the change in geometry brought about by ice flow.





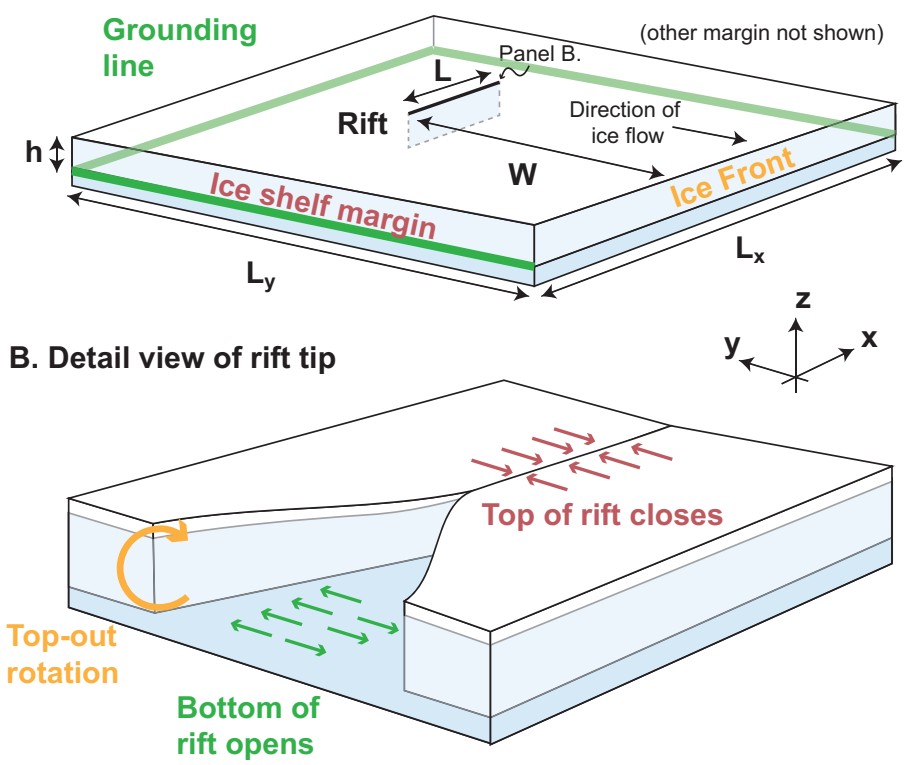

**Figure 1.** A. Simplified geometry of an idealized rectangular ice shelf. B. Zoomed in view of an ice shelf rift tip showing how buoyancy-driven rotation of the rift walls results in partial contact of the rift walls near the rift tip.

Previous studies have examined the flow–fracture relationship in several ways. Hulbe et al. (2010) carried out viscous flow calculations to constrain the state of stress in their elastic calculations. They then tuned elastic moduli and boundary conditions in their elastic calculations to match the observed viscous stresses. Plate et al. (2012) parameterized a state of stress from a viscous flow model, but rather than tuning elastic moduli instead chose to introduce fictitious equivalent body forces. Here, I

consider the hypothesis that the forces that drive rift propagation are entirely described by the instantaneous ice shelf geometry and boundary conditions. This hypothesis requires three-dimensional calculations in order to directly calculate –rather than parameterize or approximate) the role of gravitational driving forces. I next describe the details of my mechanical model.





## 3 Mechanical Model

### 3.1 Geometry

I consider the idealized ice shelf geometry shown in Fig. 1. The ice shelf is square in map view (the $x$-$y$ plane). The $z$ axis is defined so that the positive $z$ axis points upwards. The ice shelf has horizontal dimensions $L_x = L_y = 100$ km and thickness $h = 200$ m. The ice shelf surface at $y = 0$ faces the ocean and the surface at $y = L_y$ faces the ice sheet. The surfaces at $x = 0$ and $x = L_x$ are referred to as the ice shelf margins. A single rift is located along the $x$ axis at $y = W$. I treat two different general rift locations: marginal and central. These two rift locations are shown in Fig. 2. I hold the rift length fixed at $L = 2.5$ km long 90 for the marginal rift and $L = 5$ km long for the central rift.

Geometrically, I model a rift as a tapered rectangular hole in the ice shelf. Fractures in three dimensions have a fracture tip defined by a two dimensional curve rather than a point. Although I refer to a rift tip for brevity, this term actually refers to a rift tip curve. In the treatment presented here, the rift tip curve is taken to be a vertical straight line. The rift is uniformly 10 m wide over most of its length. Simulations show negligible sensitivity to the choice of this width. Tapering is applied over a length 95 equal to several widths (i.e., several tens of meters) near the rift tip.

### 3.2 Linear elasticity

I solve the equations of linear, homogeneous, isotropic, static, three dimensional elasticity (Malvern, 1969). I account for an initial hydrostatic stress in a manner following Cathles (2015) wherein the equations of elasticity are solved for a perturbation stress tensor $\mathbf{T}$ defined as the total (Cauchy) stress tensor minus hydrostatic pressure. The resulting boundary conditions 100 (described below) are consistent with previous treatments of crevasse propagation in glaciers (e.g., Van der Veen, 1998). Terms reflecting the advection of prestress (Cathles, 2015, Ch. 2, Eq. II-22) are found to be unimportant and are not discussed further. My use of isotropic elasticity implies the need for two elastic constants which I take to be Young's modulus $E = 9.7$ GPa and Poisson's ratio $\nu = 0.3$, although I sometimes make use of the shear modulus $\mu \equiv E/(2 + 2\nu)$.

### 3.3 Boundary conditions

The ice front, rift walls, and top and bottom ice shelf surfaces are loaded by a depth-varying normal stress that is equal to the water pressure below the waterline and equal to zero above the waterline. These boundaries have zero applied shear stress. I write this as a single condition on the stress tensor $\mathbf{T}$,

$$\mathbf{n} \cdot (\mathbf{T} \cdot \mathbf{n}) = -p_w(z), \tag{4}$$

with unit outward pointing normal vector $\mathbf{n}$, ice shelf draft $H_w \equiv \rho/\rho_w h$, and water pressure $p_w(z)$,

$$p_w(z) \equiv \begin{cases} \rho_w g \left[ H_w - (z + w) \right] & z < H_w, \\ 0 & z \geq H_w. \end{cases} \tag{5}$$

Here, $w$ is the vertical component of the displacement vector.





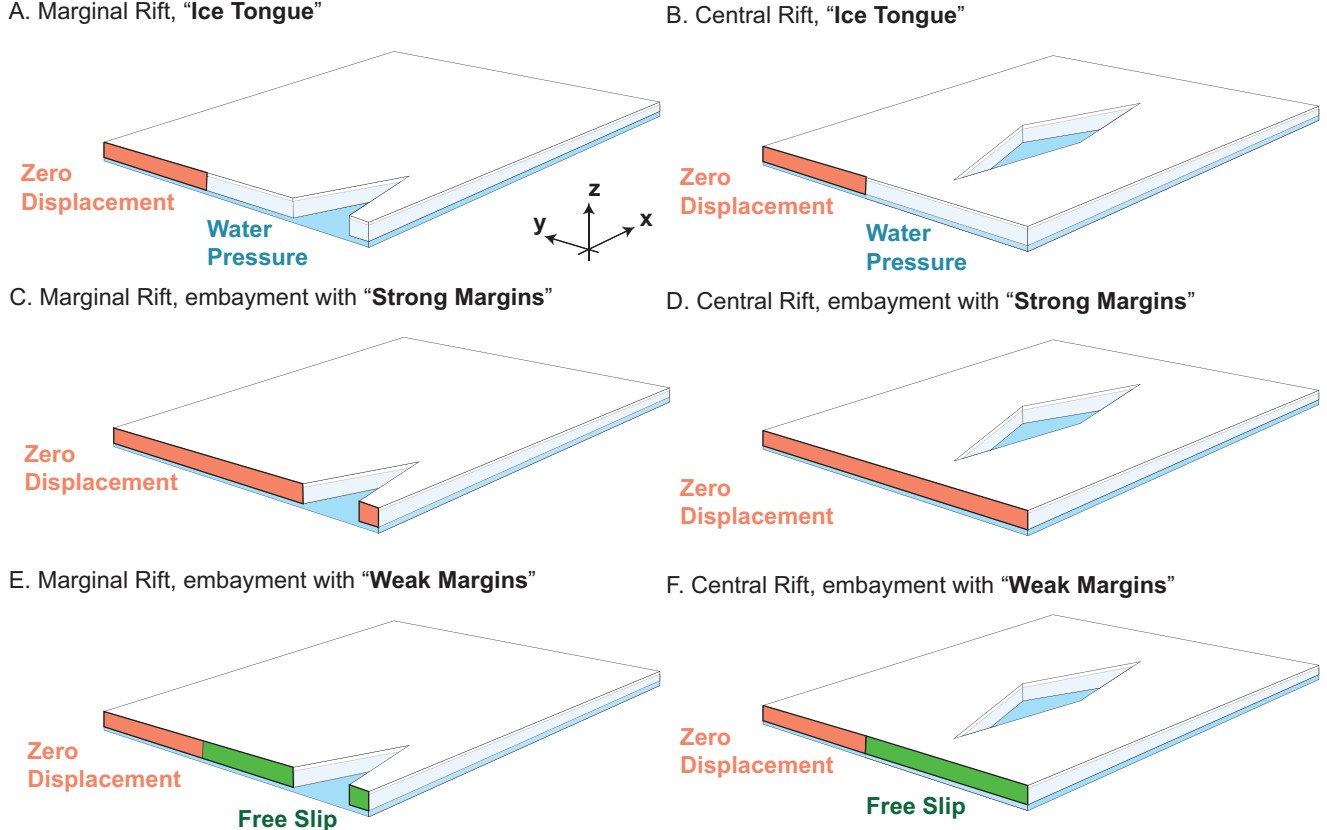

**Figure 2.** The geometries and boundary conditions considered in this study include: A. and B., Margins half zero displacement and half water pressure conditions; C. and D., Entirely zero displacement conditions; and, E. and F., half zero displacement and half free slip conditions. I furthermore consider rifts that occur in the margins (A., C., and E.) and central rifts (B., D., and F.). The figures are not drawn to scale and the rift width and shape are greatly exaggerated.

In all simulations that are presented here, the surface of the ice shelf above the grounding line at $y = L_y$ has a zero displacement boundary condition. Similarly, the ice shelf surface at the ice front at $y = 0$ has a water pressure boundary condition (Eq. 5). In the margins, I examine three types of boundary condition. These conditions are shown in Fig. 2; they are:

1. Ice shelf with ice tongue: margins have zero displacement between $y = L_y/2$ and $y = L_y$ and have water pressure between $y = 0$ and $y = L_y/2$;

2. Ice shelf in an embayment with strong margins: margins have zero displacement boundary condition; and,

3. Ice shelf in an embayment with weak margins: margins have zero displacement between $y = L_y/2$ and $y = L_y$ and have zero shear stress and zero normal displacement between $y = 0$ and $y = L_y/2$.



## 3.4  Linear elastic fracture

Cracks –including rifts– in elastic materials create displacement fields that vary proportional to the distance $r$ from the crack tip as $r^{1/2}$ (Irwin, 1957). The scalar constant of proportionality involves the stress intensity factor. Specifically, in terms of the displacement components $u$, $v$, and $w$ corresponding to displacements in the $x$, $y$, and $z$ directions, the stress intensity factors are defined through the relations (Tada et al., 2000),

$$u(r,z) = \frac{K_{II}(z)}{\mu/(2-2\nu)}\sqrt{2\pi r}, \tag{6}$$

$$v(r,z) = \frac{K_{I}(z)}{\mu/(2-2\nu)}\sqrt{2\pi r}, \tag{7}$$

$$w(r,z) = \frac{K_{III}(z)}{\mu}\sqrt{2\pi r}. \tag{8}$$

In these expressions, $r$ is the distance from the rift tip along the $x$-axis, $\mu$ is the elastic shear modulus and $\nu$ is the elastic Poisson ratio. The quantities $K_I$, $K_{II}$, and $K_{III}$ are the Mode-I, Mode-II, and Mode-III stress intensity factors. The sense of motion associated with each mode of fracture is shown in Fig. 3. Although there is also an angular dependence to the near-tip displacement fields, I have suppressed this angular dependance by writing the displacements that occur on the fracture itself.

A basic tenet of fracture mechanics is that unstable crack growth occurs when the elastic strain energy available to drive fracture exceeds the energy required to create new fracture area (Griffith, 1921). The key insight of linear elastic fracture mechanics is that this energy condition can be related to the stress intensity factors (Irwin, 1957). In three spatial dimensions, the energy release rate is (Tada et al., 2000),

$$G = \frac{K_I^2}{E/(1-\nu^2)} + \frac{K_{II}^2}{E/(1-\nu^2)} + \frac{K_{III}^2}{E/(1+\nu)} \tag{9}$$

Assuming that the Mode-I fracture toughness limits fracture behavior, we define the critical energy release rate

$$G_c \equiv \frac{K_{Ic}^2}{E/(1-\nu^2)},$$

with fracture criterion,

$$G > G_c. \tag{10}$$

## 3.5  Partial contact of rift walls

The partial contact of rift walls is a nonlinear phenomenon because it involves solving for the shape of the contacting region and therefore changing the region over which different boundary conditions are applied (Johnson and Johnson, 1987). Here, I treat a linear formulation of this problem wherein the Mode-I stress intensity factor $K_I$ can take on positive or negative values. This situation is discussed in detail by Tada et al. (2000). For fractures with zero initial width, a negative $K_I$ implies unphysical material overlap. I avoid this situation in my numerical simulations by giving the rift an initial, nonzero opening (Appendix A). This is consistent with the idea that rifts in ice shelves are probably not held open entirely by elastic stresses




because they have deformed through creeping flow. Other studies have shown that accounting for contact nonlinearity results in minimal differences from the linear problem for long fractures with $L \gg \lambda$ (Liu et al., 1999), where $\lambda$ is the ice shelf flexural

wavelength. Given that many rifts do reach lengths $L \gg \lambda$ (Walker et al., 2013, 2015), the linear approximation may well prove adequate for many cases of glaciological interest.

## 4 Results and Analysis

Figure 3 shows a typical result of the finite element calculations. This figure shows that the Mode-I and Mode-III stress intensity factors are nearly linear with depth (i.e., Fig. 3a, b, and e), while the Mode-II stress intensity factor is nearly uniform with depth

(i.e., Fig. 3d). This structure in the solutions permits an approximate parameterization of three-dimensional effects. Such a parameterization allows for a much less computationally costly, two-dimensional problem to be solved. This parameterization is developed next, in Sections 4.1. Some readers may wish to skip directly to the discussion of marginal versus central rifts in Sections 4.2 and 4.3.

### 4.1 Parameterization of 3D effects within 2D calculations

I now examine a simplified representation of the three dimensional finite element calculations that results in a parameterization for three-dimensional bending effects. The analysis hinges on the linearity of the stress intensity factors and the associated principle of superposition. I first show that the Mode-I stress intensity factor is a superposition of bending and membrane loads, whereas the Mode-II and Mode-III stress intensity factors are entirely due to membrane loading and bending loads, respectively. The stress intensity factors may therefore be approximated as,

$$K_I(z) = K_I^m + K_I^b \left( \frac{z - h/2}{h/2} \right), \tag{11}$$

$$K_{II}(z) = K_{II}^m, \tag{12}$$

$$K_{III}(z) = K_{III}^b \left( \frac{z - h/2}{h/2} \right), \tag{13}$$

where the superscripts $b$ and $m$ stand for bending and membrane, respectively.

I take the following approach to approximating the total, depth-dependent stress intensity factors. The bending terms $K_I^b$ and

$K_{III}^b$ are calculated from analytical solutions, discussed below. The membrane terms $K_I^m$ and $K_{II}^m$, in contrast, are represented in terms of geometrical functions that directly reflect the finite element solution. These two approaches are discussed in greater detail in the following two subsections. The final result of Equations 11- 13 are compared to full three-dimensional finite element solutions in Fig. 3.





### 4.1.1 The bending components of fracture

I find that the bending component of the Mode-I stress intensity factor is well fit by the simplified model (Ang et al., 1963; Sih and Setzer, 1964; Folias, 1970; Sih, 2012),

$$K_I^b = -\sigma_b f(\nu)\sqrt{\lambda}. \tag{14}$$

Here, $\lambda^4 \equiv D/(\rho g)$ is the flexural length with flexural rigidity $D \equiv Eh^3/[12(1-\nu^2)]$, Youngs modulus $E$, and Poisson ratio $\nu$. Hence, $K_I^b \sim h^{11/8}$. The bending stress $\sigma_b$ is given by Eq. (3). The function $f(\nu)$ is discussed below. Notably, the bending

stress intensity factors asymptotically vary with $\sqrt{\lambda}$ instead of the typical $\sqrt{L}$.

  There is some discrepancy in the literature concerning the precise values of the function $f(\nu)$. Sih (2012) cites Folias (1970) who both note that $f$ is of order unity but do not give its exact form. Ang et al. (1963) appears to have first given the dependence of $f$ on $\nu$ although Sih and Setzer (1964) found a mistake in this work. Meanwhile, Bažant (1992) gives a different value of $f$. It appears, however, that Bažant (1992) did not correctly account for the rift-wall boundary condition. Given this uncertainty and

the additional detail involved in the three dimensional problem beyond the assumptions made by the above authors, I instead simply choose to calculate the value of $f(\nu)$ from the three dimensional calculations. From these calculations, I find a value $f(\nu = 0.3) = 0.7646$. Of the above references, this value is most similar to the value calculated from the equation given by Sih and Setzer (1964), $f(\nu = 0.3) = 0.6063$.

  Bending also creates a Mode-III stress intensity factor. Assuming that this bending can also be described within Euler beam

theory, the Mode-III and Mode-I stress intensity factors are related by a factor,

$$\frac{K_{III}^b}{K_I^b} = \frac{h}{2\sqrt{2}(1+\nu)\lambda}. \tag{15}$$

Thus $K_{III}^b \sim h^2$, which is a larger exponent than for $K_I^b$. This solution was derived by assuming, consistent with Equations 6-8, that the ratio of stress intensity factors is proportional to the ratio of the stresses. This stress ratio is then calculated using the solution to the floating beam equation (Hetényi, 1971), $w = -2m_0/(\rho g\lambda^2)\exp(-x/\lambda)(\cos y/\lambda - \sin y/\lambda)$. The analytical

solution of Eq. (15) is compared to the finite element solution in Fig. 3e (red dashed lines).

  The analytical solution is not expected to perfectly match the finite element solution because the latter accounts for the full floatation condition (Eq. 4), whereas the bending model (Eq. 14) neglects higher order moments through Eq. (2). I further verify that the simplified model captures the behavior of the three dimensional simulations by calculating stress intensity factors over a range of ice shelf thickness between 25 m and 1600 m. I find that $K_I^b \sim h^{1.31}$ in the three dimensional calculations

whereas $K_I^b \sim h^{1.375}$ analytically. Similarly, $K_{III}^b/K_I^b \sim h^{0.27}$ in the three dimensional calculations whereas $K_{III}^b/K_I^b \sim h^{0.375}$ analytically. As can be seen in Fig. 3, the differences are more pronounced for $K_{III}^b$. I attribute the differences between analysis and calculation to the neglect of higher order moments and stress terms (i.e., the use of Euler beam theory).





### 4.1.2    The membrane components of fracture

In order to capture geometrical effects in a generic way, I introduce the following non-dimensionalization to describe the

membrane modes of fracture. I define the geometrical factors $\chi$ and $\psi$, through the relationship

$$\chi = \frac{K_I^m}{\sigma_m \sqrt{\pi L}}, \tag{16}$$

$$\psi = \frac{K_{II}^m}{\sigma_m \sqrt{\pi L}}, \tag{17}$$

where $\sigma_m$ is the depth-integrated boundary condition given in Eq. (1). In this expression $K_I^m$ and $K_{II}^m$ are calculated as

the depth-average of the finite element solution. Two comments are necessary about $\chi$ and $\psi$. First, the approximate depth-

independence of the membrane components of fracture suggests that these quantities may be calculated in simplified, two-

dimensional elasticity simulations. Verification of the two-dimensional approximation is presented in Appendix A. Second,

although the expressions in Equations 16 and 17 depend on the rift length $L$, the values $\chi$ and $\psi$ do not depend on $L$. This is

because $K_I^m$ and $K_{II}^m$ are expected to have a $\sqrt{L}$ dependence (Tada et al., 2000), therefore giving no net dependence on $L$.

### 4.2    Marginal rifts

Rifts originating in the ice shelf margins are examined in Fig. 4, which plots the stress intensity factors $K_I(z = h)$ and $K_{II}(z =$

$h)$ as a function of rift position for each of the three types of boundary conditions. Marginal ice shelf rifts become unstable in

the opening mode as they pass into the region with weak margins or onto an ice tongue (at a position $\alpha = 0.5$). The similarity

of the ice tongue and weak margin scenarios suggests that margin shear stress and not margin normal stress is the critical factor

in determining the energy release rate and hence rift stability.

Both the weak margin and ice tongue scenarios give rise to a compressive arch. The compressive arch is defined as the region

where an ice shelf transitions from uniaxial to biaxial extension (Doake et al., 1998). The compressive arch can be visualized

by plotting the second principle horizontal strain field, the first principle strain alway being positive (Fig. 5). The specific

simulation plotted in Fig. 5 is for the weak margins geometry (yellow curve in Fig. 4) with a rift located at $W/L_y = 0.57$. This

is smallest distance from the ice front where rifts are stable. The compressive arch varies in position between $W/L_y = 0.50$

near the margins and $W/L_y = 0.62$ near the center. The stability condition is therefore found to approximately relate to the

position of the compressive arch, with the exact stability threshold occurring before the rift actually crosses the arch.

Although the Mode-I stress intensity factor changes from stable $K_I < K_c$ to unstable $K_I > K_c$ as a function of rift position

$\alpha$, the Mode-II stress intensity factor has a different interpretation. This shearing mode is unstable when $|K_{II}| > K_c$ and the

sign of $K_{II}$ simply indicates the direction of shearing. With this interpretation in mind, the results in Fig. 4 suggest that marginal

rifts –anywhere in the ice shelf– are unstable in the shearing mode (i.e., $K_{II} > K_c$ for all $\alpha$) The most likely explanation for

this apparent instability is simply that, although the rift is unstable in the prescribed orthogonal geometry, upon a small amount

of propagation it will followed a curved path so as to minimize $K_{II}$. This point is discussed in more detail in the Discussion

section.

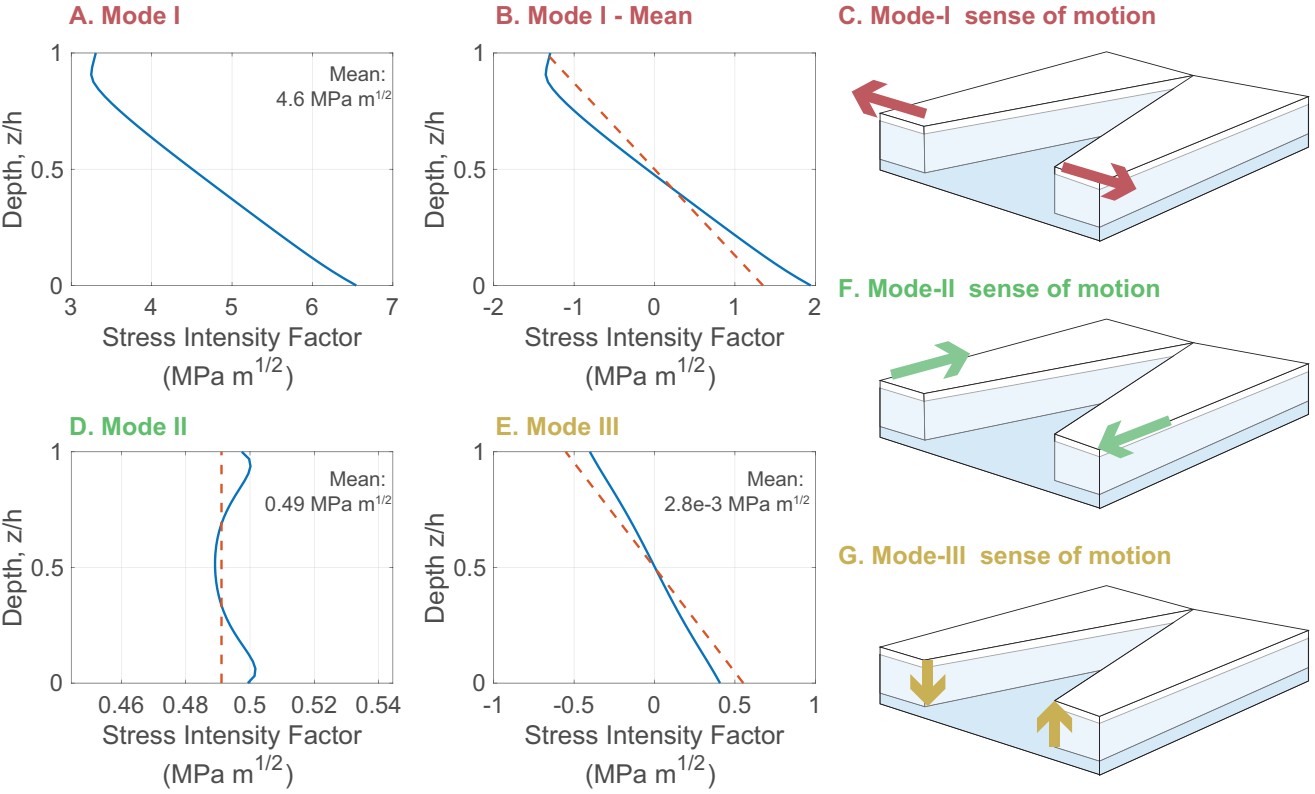

**Figure 3.** Typical three-dimensional stress intensity factors as a function of depth $z$ in the ice shelf. A. and B. show the Mode I stress intensity factor $K_I$, D. shows the Mode II stress intensity factor $K_{II}$ and E. shows the Mode III stress intensity factor $K_{III}$. The associated sense of motion for each mode is shown in panels C, F, and G. B. has the mean removed and is compared to the analytical solution of Eq. (14). The geometrical parameters for this simulation are given in the text.

## 4.3 Central Rifts

Rifts originating in the center of an ice shelf are examined in Fig. 6, which also plots the stress intensity factors $K_I(z = h)$ and $K_{II}(z = h)$ as a function of rift position for each of the three types of boundary conditions. In contrast to the marginal rifts, central rifts in all positions and with all boundary conditions are found to have negative $K_I$ and far smaller $K_{II}$, therefore suggesting the stability of this configuration. Although values of $K_{II}$ approach $K_c$, given the uncertainty in values of the fracture toughness $K_c$ (Rist et al., 2002) and given the order of magnitude greater stress intensity factors for marginal rifts, I

therefore do not interpret this as a common or significant source of instability. This point is also discussed at greater length in the next Section.



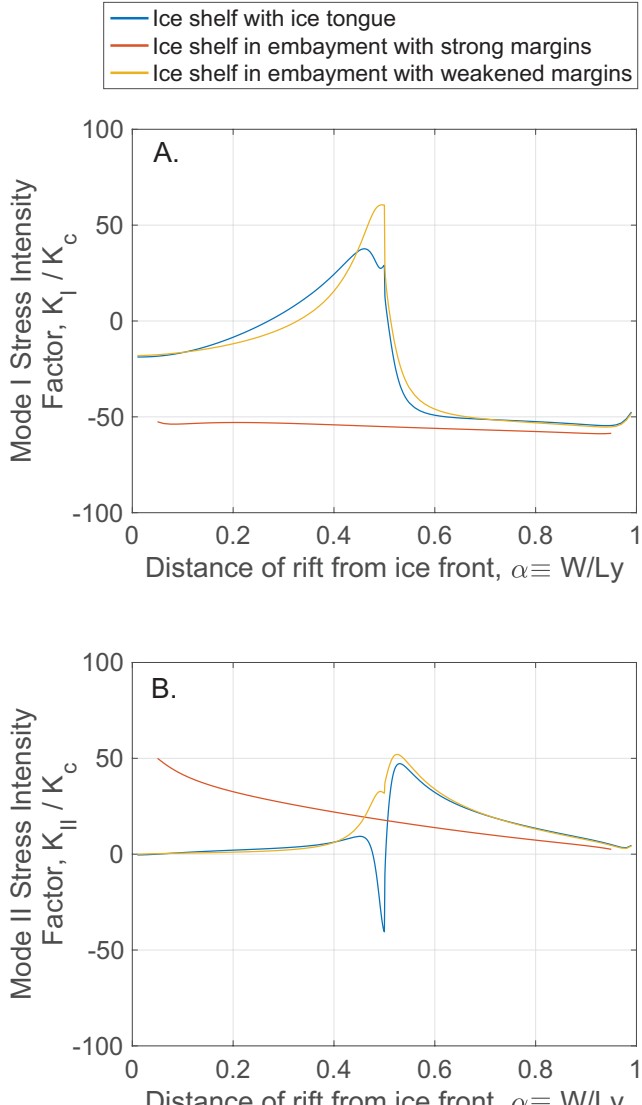

**Figure 4.** Marginal rifts in ice shelves may be either stable or unstable depending on their position and on the marginal boundary conditions. This figure plots $K_I/K_c$ in panel A and $K_{II}/K_c$ in panel B. $K_{III}$ is expected to not vary spatially and is therefore not plotted. All values are evaluated at the surface of the ice shelf $z = h$.

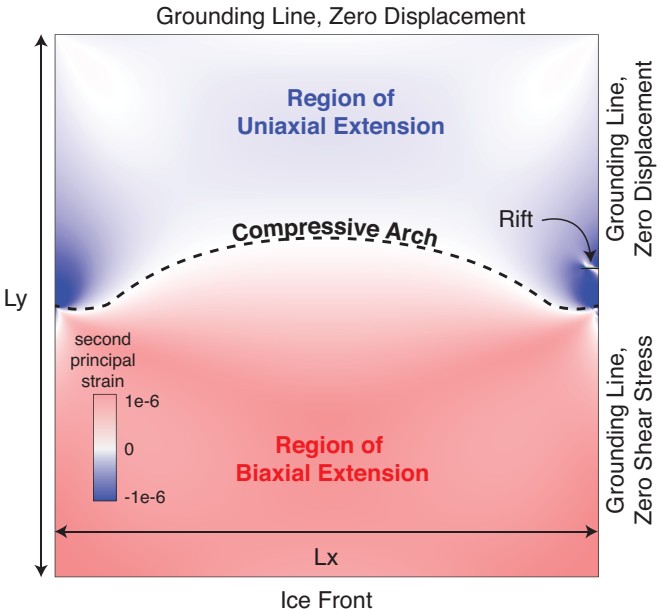

**Figure 5.** The ice shelf compressive arch (dashed line) distinguishes regions of uniaxial and biaxial extension. This occurs as the boundary where the second principal strain experiences a zero crossing. The rift location shown in this figure is in the closest stable distance to the ice front, which for a square ice shelf is calculated to be $W/L_y = 0.57$. If this rift were shifted slightly closer to the ice front it would grow in an unstable manner.

## 5 Discussion

I have presented a three-dimensional LEFM analysis of ice shelf rift propagation. The main result of this analysis is that rifts originating in the margins of ice shelves become unstable if the ice shelf margin looses shear strength. This transition between
a strong margin and a weak margin can be seen, for example, by comparing the red and yellow curves in Fig. 4. Although this result is justified by the calculations presented in this paper, it is worth emphasizing several implicit and subtle assumptions. I have assumed that margins have either zero displacement or zero shear stress. In reality, margins likely experienced reduced but nonzero shear stress. I have also considered only two rift locations (marginal or central), only one ice shelf geometry (square), and only one rift geometry (a single rift, perpendicular to flow, and without curvature). Each of these assumptions
deserves further examination. In particular, I anticipate that future work will examine the path along which fractures will tend to propagate (Erdogan and Sih, 1963; Hulbe et al., 2010; Levermann et al., 2012). Nevertheless, ice shelves and ice shelf rifts oftentimes approximately conform to these assumptions, and so I expect that the results presented here are a useful starting point in understanding additional aspects of rift propagation.





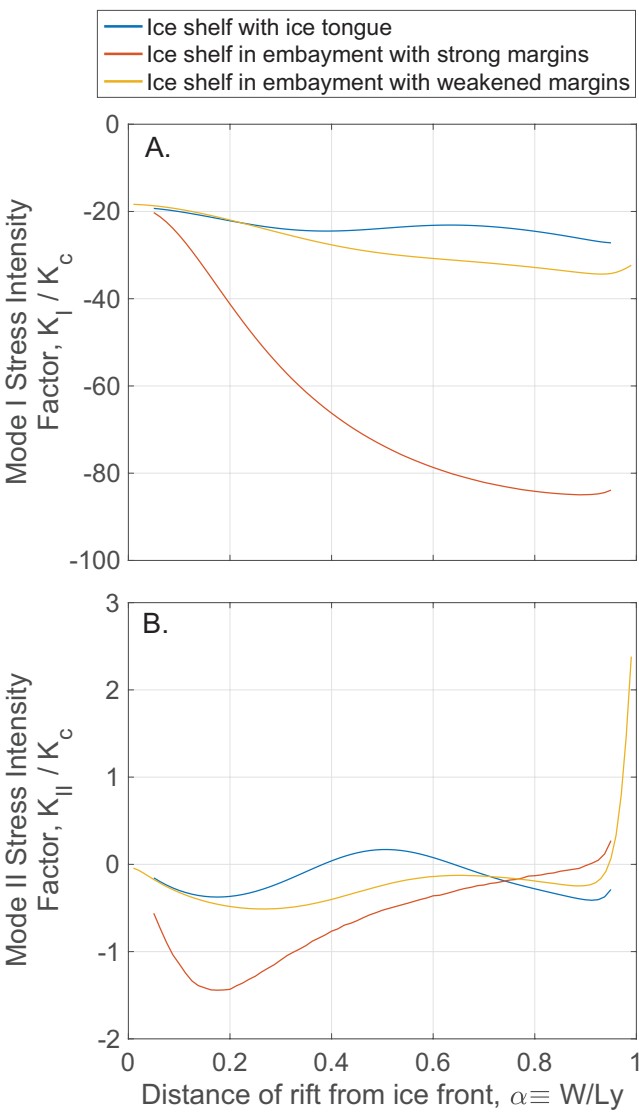

**Figure 6.** Central rifts in ice shelves are always found to be stable, regardless of their position or the marginal boundary conditions. This figure plots $K_I/K_c$ in panel A and $K_{II}/K_c$ in panel B. $K_{III}$ is expected to not vary spatially and is therefore not plotted. All values are evaluated at the surface of the ice shelf $z = h$.





### 5.1 A simplified rift propagation criterion based on flexural stabilization

The results in Section 4 suggested the approximations $G_c/G \approx 0$ and $K_{III}, K_{II} \ll K_I$. I combine these approximations with the general fracture criterion (Eq. 10) and the components of $K_I$ (Eq. 11), and evaluate the result at $z = h$, which is expected to be the most difficult part of the rift to break. The resulting fracture criterion is,

$$\chi \sigma_m \sqrt{\pi L} \geq \sigma_b f(\nu) \sqrt{\lambda}.$$

This condition has the interpretation that flexurally-induced partial contact of rift walls acts in a manner similar to the fracture toughness $K_c$ insofar as it is a resistance to rift growth that does not depend on rift length $L$. I refer to this phenomenon as flexural stabilization. This condition may be further simplified by using Equations 1 and 3,

$$L \geq \begin{cases} \chi^{-2} F(\nu, \rho/\rho_w) \lambda \equiv L_c & \chi > 0, \\ \infty & \chi < 0, \end{cases} \tag{18}$$

where $F$ is a function that depends only on Poisson's ratio $\nu$ and $\rho/\rho_w$. For $\nu = 0.3$ and $\rho/\rho_w = 0.89$, $F = 0.0716$. The expression of Eq. (18) defines the critical rift length for stability $L_c$. It has the interpretation that rifts are expected to grow unstably when they exceed a length $L_c$ proportional to the flexural length $\lambda$. For the case shown in Fig. 3, $\chi = 0.55$, giving $L_c = 217$ m. However, given such a small $L_c$, in many cases of practical glaciological importance $\chi > 0$ may well approximate the criterion for rift propagation.

### 5.2 Melange as a rift proppant

Olinger et al. (2019) observed a lack of rift-tip seismicity at central rift in the Ross Ice Shelf. This observation is consistent with the negative $K_I$ I have calculated for centrally-located rifts. In the absence of other forces such rifts will tend to close. It seems likely that these rifts therefore owe their continued existence to rift-filling melange that acts as a type of proppant by holding the rift open. Melange therefore has a dual nature. MacAyeal et al. (1998) and Rignot and MacAyeal (1998) showed that melange maintains shear stresses and therefore resists viscous flow. In this sense, melange is stabilizing. Yet in the sense that melange may sometimes enable the existence of rifts that would otherwise close, melange is destabilizing.

### 5.3 Wave-induced fracture

Lipovsky (2018) used passive seismic data to calculate the elastic ice shelf stresses due to ocean swell acting on the Ross ice shelf. This study concluded that some un-modeled process must have been operating in order to explain the lack of any observed ice shelf rift propagation during the observation period. Specifically, Lipovsky (2018) calculated a maximum wave-induced Mode-I stress intensity factor $K_I \approx 2$ MPa m$^{1/2}$ for a site near the Nascent Iceberg Rift. Using the results presented here for a central rift, we calculate that for a near-front central rift with $W/L_y = 0.05$, the Mode-I stress intensity without wave stress would be $K_I \approx -5$ MPa m$^{1/2}$. The resulting total Mode-I stress intensity factor of $K_I \approx -3$ MPa m$^{1/2}$ being negative is consistent with the observation that ocean swell did not trigger rift propagation during the observation period described by Lipovsky (2018).





## 5.4 The compressive arch

285 Doake et al. (1998) proposed that "once a retreating ice front breaks through the critical 'compressive arch' then retreat is irreversible." The results presented here broadly confirm this hypothesis, although as shown in Section 4, the relation to the compressive arch only holds in an approximate sense. Perhaps more importantly, the results presented here suggest a slightly different order of causality than that proposed by Doake et al. (1998). Specifically, my results imply that rift propagation occurs precisely when the rift breaks through the compressive arch. Rifts on the landward side of the compressive arch, in contrast,

290 are not expected to experience propagation. Ice front retreat following is therefore expected to be irreversible as hypothesized by Doake et al. (1998). Future marginal strengthening, however, would cause a seaward migration of the compressive arch, therefore permitting a certain notion of reversibility. Future work will investigate the propagation paths taken by ice shelf rifts with respect to the compressive arch.

## 6 Conclusion

295 I have modeled an ice shelf as a three-dimensional buoyantly floating elastic plate. The resulting calculations show that through-cutting ice shelf rifts become unstable in the presence of marginal weakening or upon exiting an embayment. These results are a step towards prognostic ice shelf modeling with a physics-based relationship between ice dynamics and an ice front extent set by rift propagation.





**Table A1.** Comparison between 2D and 3D calculations

|       | $h$    | 2D       | 3D      | $\Delta$ | (2D-3D)/3D |
|-------|--------|----------|---------|----------|------------|
| $\chi$ | 100 m | -0.2937  | -0.3127 | 12.5 m   | -6.1%      |
|       | 200 m  | -0.2937  | -0.3012 | 5 km     | -2.5%      |
|       | 200 m  | -0.2937  | -0.3069 | 12.5 m   | -4.3%      |
| $\psi$ | 100 m | -0.04408 | -0.0385 | 12.5 m   | +14.5%     |
|       | 200 m  | -0.04408 | -0.0382 | 5 km     | +15.4%     |
|       | 200 m  | -0.04408 | -0.0379 | 12.5 m   | +16.3%     |

*Code and data availability.* The analysis code used in the text is available on the author's GitHub repository.

# Appendix A: Numerical implementation

## A1 Discretization

The ice shelf domain is discretized using a free tetrahedral mesh in three spatial dimensions or a free triangular mesh in two spatial dimensions. In the three dimensional simulations, the maximum element size along the rift is set to be $h/16$. The element size then increases away from the rift to a maximum value of 3.5 km. The rift is geometrically formed as a rectangular prism with width $W_{\text{rift}} = 10$ m and length $L$. I have verified that the results presented here have virtually no dependence on the choice of $W_{\text{rift}}$, provided that the value is chosen to be sufficiently small. In the two dimensional simulations (described below), the maximum element size along the rift is $W_{\text{rift}}/10$.

## A2 Stress intensity factor calculations

Stress intensity factors are directly evaluated using the asymptotic solution of Equations 6- 8. This evaluation method is sometimes called the displacement correlation method (Zehnder, 2012) and has previously been used in glacier studies by Jimenez and Duddu (2018). In three dimensions, displacement differences across the rift are calculated at various heights through the ice shelf thickness, with the resulting calculations plotted in Fig. 3.

The uniformity of $K_I^m$ and $K_{II}^m$ with depth permits simplified two-dimensional elasticity calculations of these quantities. In two dimensions, displacement differences across the rift are calculated at a single set of points. A comparison of the geometrical parameters $\chi$ and $\psi$ calculated in two and three spatial dimensions is presented in Table A1. The agreement is better for $\chi$ than for $\psi$, with differences on the order of several percent.

*Competing interests.* The author declares that no competing interests are present.



*Acknowledgements.* This project began with a discussion with Colin Meyer at the Cambridge symposium of the International Glaciological Society in 2015. Jim Rice and Eric Dunham read earlier versus of this manuscript and gave the author helpful comments. Several discussions

with Brent Minchew, Jan De Rydt, and Hilmar Gudmundsson also helped along the way. On a visit to C.U. Boulder organized by Jed Brown, David Marshall was critical of some early results; this feedback was helpful. The author was funded by the Department of Earth and Planetary Sciences at Harvard University.



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
