# Peer review of "Ice shelf rift propagation: stability, three-dimensional effects, and the role of marginal weakening"

_The Cryosphere, 2019_

## Referee Comment (RC1) · Anonymous Referee #1 · 25 Nov 2019

General comments:

This is an extremely timely and creative paper that approaches a physical process that many glaciologists would be timid to approach in such a straightforward manner. Bravo!

The manuscript is well written, and I have only indicated a few suggestions as for improvements. (One place where I was confused was caused by not realizing the nature of how the solutions were derived, numerically using Equations 6, 7 and 8, apparently, as stated in the appendix. I didn't at first notice a reference to the appendix in the text.... after searching for it, I see it is referenced on page 7. It may be that the main body of the text needs a more forthright statement about what is done to produce the results, and a "louder" statement of what Appendix A is about would be helpful to some readers.)

[Figure]

Interactive
comment

I think it is important to state somewhere at the outset and also in regard to future research that the fern-structure of the ice shelf may have additional bearing on the problem. In this case, the "touching of the top" is by weak, crushable firn. Also parts of the ice shelf that are in snow accumulation areas will have rift tops that are being actively filled with new material. Dealing with this is far beyond the scope of the present paper, but it worth identifying as a factor in future investigation.

I am very impressed with the fact that observations, specifically (1) the absence of seismicity at rift tips, (2) the failure of a wave-forced propagation of a Nascent rift, and (3) the view of the compressive arch by Doake, are so nicely explained by the simple analysis of the theory presented. This, to me, is a great success and one which suggests that this approach may be what breaks any "log jam" over how rifting on ice shelves is to be pursued in the future.

Specific comments:

Around line 30 of page 2. I wonder if a citation to a paper by Sanderson would be appropriate. He thought about ice-shelf margins. Journal of Glaciology, V22, 1979.

Is equation 2 the bending moment due to the stress balance at the ice front that leads to a bending moment? Just a comment would suffice.

In the discussion along with Figure 1, it may be useful to point the reader to observational studies of rift walls: e.g., Scambos, T., Ross, R., Haran, T., Bauer, R., Ainley, D., Seo, K., . . . MacAyeal, D. (2013). A camera and multisensor automated station design for polar physical and biological systems monitoring: AMIGOS. Journal of Glaciology, 59(214), 303-314. doi:10.3189/2013JoG12J170 Note figure 8 in that paper.

In describing the model, I think it is important to state whether a firn layer is going to be treated or not. Also, although a minor point: I wonder if it is worth mentioning that brine-infiltration, horizontally along the bottom of the firn where it is permeable and where there is an ice front or rift wall might introduce secondary effects on rift wall
bending moments etc..

page 9 just below line 185. f is given to 4 significant digits. I wonder if this could be considered misleading. I also note that the Young's modulus that is used in the study is expressed as if it were very accurately known. My understanding is that relative sizes are more likely to be significant in terms of what readers take away from the comparison t this point in the paper. Perhaps that should be stated.

Figure 5, and some of the preceding figures. Do these results present the solution of Equations 6 7 and 8? I'm confused as to the specific process required to generate the curves and 2-d plot of displacement and other factors. A simple summary (before the results are presented) that describes how the model is implemented would be helpful to other researchers. Oh Dear! I see that this is all explained in the Appendix. (I should have noticed!) But, if my confusion (missing the reference to the appendix) can be of service in improving the exposition, let it so be.
* * *

---

## Referee Comment (RC2) · Anonymous Referee #2 · 17 Jan 2020

The paper "Ice shelf rift propagation: stability, three dimensional effects, and the role of marginal weakening" by Bradley Paul Lipovsky presents a study on the stability of central or marginal rifts for different boundary conditions. The stress intensity factors are computed with the displacement field from a three-dimensional finite element model assuming linear elasticity. These factors are then used as a criterion if the rift is stable (critical stress intensity factor is larger than the computed stress intensity factor) or not. Marginal weakening by free slip or water pressure destabilizes rifts that originate in the margins and central rifts are always found to be stable.

General comments: - What is the value the author uses for the critical stress intensity factor $K_c$? In the text, there is no explicit value given.

- Could the author please include the text of both appendices in the main text? The

points that are discussed there are critical for the comprehension of the paper. There is no reference to the Appendix and it is not directly clear, for instance, why the author uses the displacement in one direction only dependent on one critical stress factor of a certain loading mode (displacement direction method). The author can also shortly discuss that this method/approximation has first-order accuracy.

- At the moment it is also not clear which equations the author uses for the numerical finite element and which only for the analytical solution. For instance, Eqs. (1)-(3) are only used for the analytical solution. Is this right? For the numerical solution, the displacement field can be derived with three-dimensional elasticity and the different boundary conditions and then in a post-processing step the stress intensity factors are computed out of the displacement field. Then the author should mention this procedure in the text that it is directly clear for the reader. Maybe it is then also better to solve Eqs. (6)-(8) for the stress intensity factors: $KII(z)=\ldots$.

- Does the author also consider rifts that are not filled by water? The water cannot percolate in all rifts occurring in an ice shelf, for example, if the rift is too far away from the ice front also dry (not filled by ocean or melt water) rifts can exist. How is the stability of dry rifts? Maybe the author can also add a short comment on these studies in the text.

- Figure 3: the arrows for mode II should also be plotted at the rift edges as the author did it for mode I and mode III.

- What is the minimum element size along the rift? Did the author a mesh convergence study to also verify that the results are not mesh dependent (a crucial check if one would consider stress intensity factors at the crack tip).

Specific comments and questions: - Eq. 6-8: I do not have access to the Tada et al. 2000 paper, but are the factors in these equations right? I found in Gupta et al. 2017 ("Accuracy and Robustness of Stress Intensity Factor Extraction Methods for the Generalized/eXtended Finite Element Method") sqrt(r/(2*pi)) and mu/(4-4*nu). Could

the author please check the equations.

- Equations: Why do the author sometimes use an equality sign and sometimes the sign for identical statements with three strokes above the other (see Eqs. (2) and (3))? For example, Eq. (1) and Eq. (4) are both statements how the stress component or the stress tensor for the boundary condition is computed.

l.58: the traction boundary condition should be zero (stress-free boundary due to zero pressure) at the top of the ice shelf. The author only gets non-zero values as the simplified assumptions of Weertman and Reeh are used. Here, a comment that these results are not necessary for the finite element formulation could be helpful for the understanding of this paper.

l.93 and Fig. 2: The geometry of the rift in the figure looks like a rhomb, but in the text the rift is described with a uniformly 10 m width and only near the rift tip it is tapered. Could the author update the figure that it fits to the description of the rift? The author already states in the caption that the width and shape are exaggerated but if the author could also include the width of only 10 m in the figures, it will be clearer that LEFM could be applied where an infinitesimal small crack tip is absolutely necessary.

l. 98: Is the perturbation stress tensor the deviatoric stress tensor? Can the author also include the word deviatoric to make it directly clear for everyone and maybe add at the end of the sentence "times identity tensor"? Eq. (4): Why is the pressure boundary condition only applied for the deviatoric stress tensor and not as common to the total (Cauchy) stress tensor?

Figure 3: Could the author please add a legend to the plots A, B, D, E? Could the author please also use capital letters for the reference to the figures, see for instance l.154,155.

Why does the author choose slightly different density values of 0.9 (l.55) and 0.89 (l.263)?

l. 265: What are the boundary conditions for the case studied in Fig. 3? At each boundary water pressure? Why are all computed values of this geometrical factor negative in Table A1 (2D and 3D)? Is this due to the boundary conditions acting in an embayment? Does this statement mean that a rift longer than 217m will never be stable in a free-floating ice shelf?

Fig. 1: The author should add in the caption of this fuigure that the boundary conditions on the side of the ice shelf are too far away to have an influence on the rift. If this is not the case, then the bending moment of the water pressure at the side counteract the closure of the rift top by the opening of the rift.

Fig. 4: Why are the orange and blue curves for alpha»0.5 not reaching or converging to the red curve? The boundary conditions of free slip or water pressure are in this case far away from the rift and therefore the difference of all three cases should be minimal.

Fig. 6B: Shows the red curve in this figure not an unstable rift if alpha is in between 0.1 and 0.3? For the stress intensity factor of the shearing mode (Mode-II) it is sufficient for rift propagation that the magnitude is greater than 1 (cf. l. 228).

Technical corrections: title, l. 97, ….: standardize the spelling of three-dimensional (cf. l.3) with hyphen l.2: sea-level rise Fig. 2: Adjust the case sensitivity after a semi-colon for "Entirely" and "half" l.128: delete that mue is the elastic shear modulus and nu is the elastic Poisson ratio as both of them are already introduced in l. 103 l.131: dependence instead of dependence l.169: either "approach to approximate" or "approach approximating" l.172: add a s to results: "the final results of Eqs. (11)- (13) are compared" Equations: They should be referred to by the abbreviation "Eqs." and the respective numbers in parentheses, see l. 172, l. 261, l.309

---

## Author Comment (AC1) · 28 Jan 2020

**Response to Reviewer #1**

One place where I was confused was caused by not realizing the nature of how the solutions were derived, numerically using Equations 6, 7 and 8, apparently, as stated in the appendix. I didn't at first notice a reference to the appendix in the text.... after searching for it, I see it is referenced on page 7. It may be that the main body of the text needs a more forthright statement about what is done to produce the results, and a "louder" statement of what Appendix A is about would be helpful to some readers.

In a revised manuscript, I have made a significant effort to address this concern. I have added text at the beginning of Section 3 that clarifies the modeling approach. I've also incorporated the Appendix into the main text in a way that is more clear for the reader. I have also reorganized subsections (and sub-subsections) in Sections 3 and 4 so that the paper outline offers a better guide of the calculations that are presented. I have also added several "signposts" throughout the manuscript that attempt to orient the reader.

I think it is important to state somewhere at the outset and also in regard to future research that the fern-structure of the ice shelf may have additional bearing on the problem. In this case, the "touching of the top" is by weak, crushable firn. Also parts of the ice shelf that are in snow accumulation areas will have rift tops that are being actively filled with new material. Dealing with this is far beyond the scope of the present paper, but it worth identifying as a factor in future investigation.

Agreed. In a revised manuscript I have now explicitly mentioned the role of firn and its approximation in my model. I found the most natural place for this clarification to be at the beginning of the discussion section.

I am very impressed with the fact that observations, specifically (1) the absence of seismicity at rift tips, (2) the failure of a wave-forced propagation of a Nascent rift, and (3) the view of the compressive arch by Doake, are so nicely explained by the simple analysis of the theory presented. This, to me, is a great success and one which suggests that this approach may be what breaks any "log jam" over how rifting on ice shelves is to be pursued in the future.

Thank you!

Around line 30 of page 2. I wonder if a citation to a paper by Sanderson would be appropriate. He thought about ice-shelf margins. Journal of Glaciology, V22, 1979.

Sanderson was previously cited in my manuscript but not on this point. I have added this reference at this location as suggested.

Is equation 2 the bending moment due to the stress balance at the ice front that leads to a bending moment? Just a comment would suffice.

Yes, that's right. I've tried explaining this slightly differently in the text.

In the discussion along with Figure 1, it may be useful to point the reader to observational studies of rift walls: e.g., Scambos, T., Ross, R., Haran, T., Bauer, R., Ainley, D., Seo, K., . . . MacAyeal, D. (2013). A camera and multisensor automated station design for polar physical and biological systems monitoring: AMIGOS. Journal of Glaciology, 59(214), 303-314. doi:10.3189/2013JoG12J170 Note figure 8 in that paper.

I'm grateful to the reviewer for pointing out this reference. I've added a mention to it.

In describing the model, I think it is important to state whether a firn layer is going to be treated or not.

I do believe that treating the firn layer is beyond the scope of the present paper. Since the firn layer may nevertheless be of importance, I have added text stating this point at the beginning of Section 3.1 as well as in the Discussion section.

Also, although a minor point: I wonder if it is worth mentioning that brine-infiltration, horizontally along the bottom of the firn where it is permeable and where there is an ice front or rift wall might introduce secondary effects on rift wall bending moments etc..

I have made additional note of this process (start of Section 5).

page 9 just below line 185. f is given to 4 significant digits. I wonder if this could be considered misleading. I also note that the Young's modulus that is used in the study is expressed as if it were very accurately known. My understanding is that relative sizes are more likely to be significant in terms of what readers take away from the comparison t this point in the paper. Perhaps that should be stated.

I have changed this mistake. Significant digits are now consistently reported.

Figure 5, and some of the preceding figures. Do these results present the solution of Equations 6 7 and 8? I'm confused as to the specific process required to generate the curves and 2-d plot of displacement and other factors. A simple summary (before the results are presented) that describes how the model is implemented would be helpful to other researchers. Oh Dear! I see that this is all explained in the Appendix. (I should have noticed!) But, if my confusion (missing the reference to the appendix) can be of service in improving the exposition, let it so be.

Yes, this is useful.  Again, as noted in the first comment above, I have made a significant effort to clarify the exposition.

**Response to Reviewer #2**

- What is the value the author uses for the critical stress intensity factor Kc? In the text, there is no explicit value given.

Following the work of Rist et al (2002), I use 100kPa m^{1/2}. None of the results in the manuscript are sensitive to this precise value. I have added text at the end of Section 3.2 that provides this information.

- Could the author please include the text of both appendices in the main text? The points that are discussed there are critical for the comprehension of the paper.

Yes, I have moved the text in the Appendix to the main text.

There is no reference to the Appendix and it is not directly clear, for instance, why the author uses the displacement in one direction only dependent on one critical stress factor of a certain loading mode (displacement direction method).

It is simply a matter of definition that each mode depends on an orthogonal component of displacement. This definition, however, bears utility in its relationship to fracture propagation. I have modified the text in Section 3.2 following the definitions of the stress intensity factors to explain this point.

The author can also shortly discuss that this method/approximation has first-order accuracy.

I have made such a note.

- At the moment it is also not clear which equations the author uses for the numerical finite element and which only for the analytical solution.

Following on the comments from Reviewer #1, I have made changes to the manuscript in an effort to improve clarity on exactly this point. I have added the exact equations that are solved in Section 3.1. I have also added text at the beginning of Section 3 that clarifies the modeling approach.

For instance, Eqs. (1)-(3) are only used for the analytical solution. Is this right?

Yes, and I have added a note to this extent in Section 3.1.

For the numerical solution, the displacement field can be derived with three-dimensional elasticity and the different boundary conditions and then in a post-processing step the stress

intensity factors are computed out of the displacement field. Then the author should mention this procedure in the text that it is directly clear for the reader. Maybe it is then also better to solve Eqs. (6)-(8) for the stress intensity factors: KII(z)=: : :.

Yes, that is exactly correct. I have added a note that clarifies this point in the introduction as well as at the beginning of Section 3.

- Does the author also consider rifts that are not filled by water? The water cannot percolate in all rifts occurring in an ice shelf, for example, if the rift is too far away from the ice front also dry (not filled by ocean or melt water) rifts can exist. How is the stability of dry rifts? Maybe the author can also add a short comment on these studies in the text.

This is an interesting point and I have made mention of it at the beginning of Section 5.

- Figure 3: the arrows for mode II should also be plotted at the rift edges as the author did it for mode I and mode III.

I've made this change.

- What is the minimum element size along the rift? Did the author a mesh convergence study to also verify that the results are not mesh dependent (a crucial check if one would consider stress intensity factors at the crack tip).

Yes, this is an important point.  I did verify that the results are not mesh-dependent prior to initial submission. The maximum element size near any boundary, including the rift tip, is constrained to be no greater than h/16 with ice thickness h.  Furthermore, stress intensity factors are measured over several elements, an essential aspect of mesh independence. Although some of these points were already described in the text, I have added more detail in the newly-created Section 3.2.2.

Specific comments and questions:

- Eq. 6-8: I do not have access to the Tada et al. 2000 paper, but are the factors in these equations right? I found in Gupta et al. 2017 ("Accuracy and Robustness of Stress Intensity Factor Extraction Methods for the Generalized/eXtended Finite Element Method") sqrt(r/(2*pi)) and mu/(4-4*nu). Could the author please check the equations.

I'm grateful for the reviewer's attention to detail for catching this mistake.  I verified that these equations were correctly implemented in my finite element calculation.  It appears that this mistake was entirely limited to the manuscript and the appropriate correction has been made.

- Equations: Why do the author sometimes use an equality sign and sometimes the sign for identical statements with three strokes above the other (see Eqs. (2) and (3))? For example, Eq. (1) and Eq. (4) are both statements how the stress component or the stress tensor for the boundary condition is computed.

I use the symbol with three lines to denote a definition.  I've checked that this is consistently used in all equations and also made a note for the reader.

l.58: the traction boundary condition should be zero (stress-free boundary due to zero pressure) at the top of the ice shelf. The author only gets non-zero values as the simplified assumptions of Weertman and Reeh are used. Here, a comment that these results are not necessary for the finite element formulation could be helpful for the understanding of this paper.

Yes, agreed.  I added a sentence clarifying this point in Section 3.2.1.

l.93 and Fig. 2: The geometry of the rift in the figure looks like a rhomb, but in the text the rift is described with a uniformly 10 m width and only near the rift tip it is tapered. Could the author update the figure that it fits to the description of the rift? The author already states in the caption that the width and shape are exaggerated but if the author could also include the width of only 10 m in the figures, it will be clearer that LEFM could be applied where an infinitesimal small crack tip is absolutely necessary.

I have updated Figure 2 as the reviewer suggests.

l. 98: Is the perturbation stress tensor the deviatoric stress tensor? Can the author also include the word deviatoric to make it directly clear for everyone and maybe add at the end of the sentence "times identity tensor"? Eq. (4): Why is the pressure boundary condition only applied for the deviatoric stress tensor and not as common to the total (Cauchy) stress tensor?

Briefly, no, it is not the deviatoric stress tensor.  The perturbation tensor is T- p_0 whereas the deviatoric tensor is T-p.  As this is an important point, I explain the difference in detail in the beginning of Section 3.1.  The perturbation tensor still allows for elastic compressibility whereas the deviatoric stress tensor does not.

Figure 3: Could the author please add a legend to the plots A, B, D, E? Could the author please also use capital letters for the reference to the figures, see for instance l.154,155.

I have made these changes.

Why does the author choose slightly different density values of 0.9 (l.55) and 0.89 (l.263)?

This was an oversight. In the revisions I've opted to use the value with fewer significant figures to reflect the uncertainty in this quantity.

l. 265: What are the boundary conditions for the case studied in Fig. 3? At each boundary water pressure?

I have clarified the figure caption to state that this figure is drawn for a marginal rift in a floating ice tongue.

 Why are all computed values of this geometrical factor negative in Table A1 (2D and 3D)? Is this due to the boundary conditions acting in an embayment?

Yes, this is for an embayment geometry. I have added this clarifying point to the text.

Does this statement mean that a rift longer than 217m will never be stable in a free-floating ice shelf?

This comment prompted a change to how stability is evaluated in my manuscript. In the revised manuscript, I have introduced the optimally oriented stress intensity factor. I have also removed this calculation as I do not believe it is consistent the three-dimensional results.

Fig. 1: The author should add in the caption of this fuigure that the boundary conditions on the side of the ice shelf are too far away to have an influence on the rift. If this is not the case, then the bending moment of the water pressure at the side counteract the closure of the rift top by the opening of the rift.

I have made such a note.

Fig. 4: Why are the orange and blue curves for alpha»0.5 not reaching or converging to the red curve? The boundary conditions of free slip or water pressure are in this case far away from the rift and therefore the difference of all three cases should be minimal.

This is a real effect. An essential aspect of LEFM is that distant boundaries may still alter energy release rates. This statement is epitomized by the J-Integral of Rice (1968), which expresses the energy release rate as an integral over all boundaries.

Fig. 6B: Shows the red curve in this figure not an unstable rift if alpha is in between 0.1 and 0.3? For the stress intensity factor of the shearing mode (Mode-II) it is sufficient for rift propagation that the magnitude is greater than 1 (cf. l. 228).

I have revised the previously-numbered Fig 6. It and previously-numbered Fig 4 both now show the optimally-oriented SIF. As described in the text Section 3.8, the optimally-oriented SIF provides a better measure of stability. More directly to the reviewer point, I have also modified

the discussion in (newly-numbered) Section 4.2 and 5.1 to give a more subtle description of the regimes of propagation.

**Technical corrections.** I have addressed each of the small technical corrections brought up by the reviewer.

---

## Author Response (AR2)

Dear Handling Editor Oliver Gagliardini and Anonymous Reviewers,

Thank you for the time you have invested in my manuscript. I have attempted to improve the manuscript following the comments provided below (my responses are in green).

Sincerely,

Brad Lipovsky

Response to Reviewer #3

The paper suffers however, to my point of view, of a lack of accuracy in the presentation of the model used and on the assumptions that are made. A mixture of too complicated vocabulary (and to my point of view not necessary) and not enough explanation of simple notions make the paper very difficult to follow and read.

As scientists and writers, we are often faced with the challenge of using precise and accurate vocabulary while at the same time trying to keep things as simple as possible. I've taken the reviewer's comments to heart and made an honest effort to simplify the language in the paper and to improve the structure (the latter discussed in detail further on). I believe that the exposition has significantly benefitted from these changes. I have attempted to addressed this concern throughout the manuscript by specifically responding to the following points.

Furthermore, the paper would be very much improved by adding more detailed comparison with rift propagation and ice-shelf stability in the field where I expect measurements have been made.

This is an excellent point. I have recently undertaken a systematic effort aimed at ice shelf rift observation. This is still work in progress. Unfortunately, at the present time, the extremely simplistic model geometry that we use makes it difficult to directly compare our model to observations. I therefore have the opinion that comparison with observations is best left at a relatively qualitative level. A note to this extent has been provided in the manuscript. That being said, the discussion does provide several points of comparison with ice shelf rifts. I specifically discuss observations surrounding the role of mélange as well as the seismic observations made by Lipovsky (2018). The present study has demonstrated that geometrical effects are quite important during rift propagation. For this reason, I do think that more detailed comparisons are better left for a modeling study that deals with a more realistic ice shelf geometry.

I detail below the points that have to be clarified:
0 - Abstract:
L4-5: The sentence is not clear: '… near-tip rift walls'.

I have simplified this sentence.

L6-7: It is not obvious by reading just the abstract to understand '….advection of rifts … may trigger rift propagation'. In particular what does mean 'advection of rift' ? How is it different from rift propagation? This has to be more clearly stated for readers that are not exactly in the field.

I have simplified this language.

More generally, when reading the last sentence, it seems that there is no other studies on the description of calving physics based on fracture mechanics. It seems however to me that other studies have been done in this direction.

I have changed this sentence.

1- Introduction

In the introduction, discussion and conclusion, there is a lack of detailed description of what is observed on the field related to the study performed in this paper. Furthermore, there is a lot of work based on fracture mechanics that have been done in rock mechanics and also in rupture propagation for earthquakes. Because no references related to this work are present in this paper, it seems that the author is not aware of it, making it difficult to situate his work compared to the state of the art in the domain.

I have significantly changed the ordering of ideas presented in the introduction. Some material has been moved to Section 4. As previously written, the paper introduces only fracture mechanical studies of ice shelf rifts in the introduction section. I decided not to clutter the introduction with fracture mechanical detail. The interested reader will note the extensive discussion of classical fracture mechanics later in the paper, at the point where fracture mechanics is introduced.

2 – Background
One of the main problem of this paper is that the equations and hypothesis used are not clearly presented. At the beginning of the background section, simple equations described by complicated words are presented.

I agree that this section of the paper was confusing. For this reason, I have restructured this section, mostly by moving existing text to different parts of the paper. In that way, the same information is conveyed but in a more logical order. In the new manuscript, I now make it clear that the equations previously in Section 2 are only used in the two-dimensional model. For this reason, these equations are now given in the newly-created Section 4, dedicated exclusively to describing the two-dimensional model.

First, equation (1) is related to the hydrostatic approximation for the pressure taking into account buoyancy but is referred to as 'in-plane horizontal membrane stress'. While I assume that the author refer to the shallow ice approximation, the use of such words do not provide clear description of the approximation made at least for researchers that are not specialists of the ice-shelf and rift problem.

Just to be clear, the model is not related to the shallow ice approximation (SIA) model. The two dimensional model is more similar to a shallow shelf approximation (SSA) model, but it does not assume incompressibility as does SSA. The term membrane stress is widely used in modern ice sheet modeling and its use here is consistent with that literature. I believe that the confusion surrounding this topic should be alleviated by having moved this material to a later section, as described above.

To make the considered forces clear, I suggest to draw the forces involved in Figure 1 or Figure 2.

The reviewer proposes an interesting idea to create additional diagrams illustrating the balance of forces. Because of the way that the paper has been restructured, I'm not sure that this is necessary. Furthermore, such figures are given in standard references (i.e., Macayeal). As it is currently written the manuscript contains two diagrammatic figures (1A and 1B). For these reasons, I do feel that an additional diagrammatic figure is not warranted.

Because the cryosphere readers are not all aware of the way you calculate the balance of forces, this should be clearly recalled as it is the basis of this paper.

I have added a sentence to this effect in Section 4, where the depth-integrated membrane stress is now first stated.

I don't understand why the author used the symbol    that he refers to 'a definition'. Equation (1) is not a definition, it represents the normal stress under some approximations. Again, this kind of things complicate the problem for nothing (at least I don't see what information it provides to the reader) and is very strange when put in regards to the very simple approximations, equations, and approach used here (with no demonstration, etc.).

This confused another reviewer as well, so I've gotten rid of all occurrences of the three-line equality symbol.

L 58: Demonstrate how to you express the bending moment leading to equation (3) (use a figure if necessary).

First, please note that this equation is now placed in Section 4 as Eq. 21. I am not completely sure what the reviewer means by "express." The new context in Section 4 may provide a better

expression of the meaning of this equation. The calculation of moments is carried out in the classic paper by Reeh. Given that the manuscript under consideration already on the long side, it is my opinion that explicitly presenting the derivation or figures regarding the ice front moment expressions would not significantly improve the manuscript.

L66: show in the figure 1b what you state in the text.

I have now made the language in the text match the language in the figure.

L72: There should be studies on 3D effects on rupture propagation in Earthquake or rock mechanics that should be recalled here.

For reference, the old line 72 stated, "Although a number of previous studies have examined ice shelf rifts using LEFM, no previous study appears to have considered three-dimensional effects." Note that later on in the paper, extensive reference is given to background material in fracture mechanics. Concerning the point presented here, yes, there is a large literature on 3D effects in earthquake rupture dynamics. The 3D effects in ice shelves, however, are quite different because ice shelves, as the manuscript demonstrates, have strong interactions with buoyancy. These effects are absent during, for example, tectonic earthquake rupture. Furthermore, earthquake rupture tends to involve rupture front propagation at a substantial fraction of the elastic wave speeds. Fracture propagation at this rate is drastically different than the quasi-static propagation considered here. Given that this connection is sufficiently distant, it seems preferable to not draw this connection.

Figure 1 : It is difficult to make the link between Figure 1A and Figure 1B. To avoid getting lost, the author could draw the flow direction in each sub-figures of Figure 1 and 2. The orange arrow associated to 'Top-out rotation' correspond to me to 'Bottom-out' rotation for calving. What is the point here ? Represent the rift tip on the figure. Legend of Figure 1B: I don't see on the figure what is stated in the legend 'Zoomed in view of an ice shelf rift tip showing how buoyancy driven rotation of the rift walls results in partial contact of the rift walls near the rift tip'. Define in the text or in the legend how the flexural gravity wavelength is calculated.

I have attempted to address these issues. First of all, I have added a reference to the equation where the flexural gravity wavelength is defined in the figure caption. Concerning the definition of top-out versus bottom out, I would argue that this is why it is important to define terms. This usage of the term makes sense to me: the top of the ice shelf moves outwards away from the ice. Perhaps there is some confusion because the top moves towards the center of the rift. In that way the definition is arbitrary and I would argue that as long as the usage is consistent then the choice is unimportant. The term top out is used self consistently within the manuscript and is consistent with sense of rotation drawn in the figure. Given this self-consistency I do not believe that any change is warranted.

3 – Mechanical model

What is the link between the background section and the model used here ?

This question highlights the lack of clarity surrounding the relationship between the full three-dimensional and the simplified two-dimensional model.  Based on this comment, I have opted to move part of the background material into the later section about the two dimensional model, as discussed previously.

Why two different L are chosen for the marginal and central rifts. It makes the message unclear as the role of L may be important in the different behavior of the marginal and central rifts. Could you explain your choice and could you do the analysis by comparing marginal and central rifts for the same L ?

This is a standard convention in fracture mechanics, to define fractures in a whole space in terms of their half-length but to define edge fractures in terms of their entire lengths.  This convention stems in part from analytical solutions for simplified geometries.  The usage in the manuscript is consistent with standard stress intensity factor handbooks such as Tada (2000) and for this reason I would argue against making the suggested change.

L 97-101 : This is not clear, illustrate on a figure.

The actual crack tip region used in the simulations is shown in Figure 2 and a curved crack tip is shown in Figure 1b.  A note has been made about this.

L 100: Show in the appendix the sensitivity to the choice of the width because it is not obvious how much its influence is negligible.

This is a good point.  I have added a note (in the main text, Section 2.1) describing the influence of changing the rift width.  I point out that changing the rift width by 50% results in a ~1% change in the resulting SIFS.

L 100-101: Give more details about what you do when you refer to 'tapering'.

This language was out of date; the rectangle is not tapered.

Equation (5): You forget the Identity tensor. Same L 116.

I have made this change.

Equation (6): What is H ? Is it h in Figure 1 ? Then the same notation should be used. H is not constant when there is a rift so that the horizontal pressure gradient could not be neglected when replacing T' by T ? As a result, equation (7) is not obtained. Maybe I missed something but all this should be made clearer.

Thank you for pointing this out, this symbol was not defined previously! H is the ice shelf surface height. H remains constant even though H is not defined in the rift For this reason (old) equation 7 remains valid.

Equation (10): As you makes a variable change and use T instead of T', the boundary conditions should be expressed in terms of T too.

Yes, that's correct that the boundary conditions change. All of the terms in the new boundary conditions have been written out already, and since the new boundary conditions are more complicated I do not think it is necessary to write out the new boundary conditions. It is an important point, however, and so I have made a note to this extent in order to draw the reader's attention to this detail.

L 135: relate the displacement vector to u_i, u_j, …

I have made this change.

Figure 2: The flow direction should be added here too.

I have made this change.

L 145-146: What are the boundary conditions used in the 3D calculations then ?

This sentence was confusing so I removed it. The 3d boundary conditions are now clearly stated.

L 150 : recall what is 'free' (triangular mesh)

I have clarified this point.

Equations (12)-(14) Recall the hypothesis made to obtain these equations

It's a surprising fact about LEFM that all of the conditions were in fact already stated: simply that a sharp fracture be located in a loaded elastic solid. There is nothing else to be stated.

L 200: It is not clear how do you calculate them.

On this topic, I refer the interested reader to the citations given.

L 260: It could be good to show it on a figure.

This is a minor point in the paper and I do not think it warrants an additional figure.

5 - Discussion: The main problem of the discussion is that the results are not enough compared with field observation.

Please note the response I made to the second comment in the review.  I would furthermore argue that I do compare to the field observations described by Lipovsky (2018).  These are seismic observations from which inferences were made about stress intensity factors.  I also discuss the field observations made by Olinger et al. (2019).  While I agree that additional and more detailed comparisons would be interesting, these two comparisons do form a good basis for an evaluation of the model predictions.

L 299-301: Put marks on Figure 6 to show what is stated in the text.

This was already shown in the figure although the confusion indicates that the labelling was not sufficiently clear.  I have improved the text in this regard.

L 304-305 and L 308-309 are not clear.

I have rewritten these statements.

6 - Conclusion: This should recall the assumptions made in the model and summarize the results, the limitation of the approach and the comparison with field observation.

I disagree, as this was already done at the beginning of the Discussion section and the choice of one location over the other seems purely aesthetic.

[revised manuscript text omitted]